# Low-Order Radial Modal Test and Analysis of Drive Motor Stator

Jie Li [1,2,*], Shaobo Yang [1,2], Jincai Yang [1,2], Fengqin Li [1,2], Qingqiang Zeng [1,2], Junlong Shao [1,2], Chun Chang [1,2], Nian Wu [1,2], Ying Chen [3] and Keqiang Li [2,4]

1   Powertrain R&D Institute, Chongqing Changan Automobile Co. Ltd., Chongqing 401133, China; yangsb@changan.com.cn (S.Y.); yangjc@changan.com.cn (J.Y.); lifq1@changan.com.cn (F.L.); zengqq@changan.com.cn (Q.Z.); shaojl@changan.com.cn (J.S.); changchun@changan.com.cn (C.C.); wunian@changan.com.cn (N.W.)
2   State Key Laboratory of Vehicle NVH and Safety Technology, Chongqing 401120, China; likq@mail.tsinghua.edu.cn
3   Chongqing Vehicle Test & Research Institute Co., Ltd., National Coach Quality Supervision and Test Center, Chongqing 401122, China; cjchenying@cmhk.com
4   State Key Laboratory of Automotive Safety & Energy, School of Vehicle and Mobility, Tsinghua University, Beijing 100084, China
*   Correspondence: leejay1986@changan.com.cn

**Abstract:** For a driving motor stator of EDU (Electric Drive Unit) intelligent electric transmission of a domestic plug-in hybrid electric vehicle, modal tests are performed on the motor stator with or without motor shell. Either the hammering method or the frequency sweeping method is used in the test. The modal frequencies, modal shapes, and damping ratios of the first five orders that meet the requirements of the modal confidence criterion are obtained. The influence of the motor shell on the low-order radial modal of the motor stator is discussed. The results show that similar results are obtained in the modal parameter estimation respectively using the hammering method and the frequency sweeping method. They can both be used for low-order radial modal test of the motor stator. The motor stator without shell exhibits a linear structure in the frequency domain. Each modal frequency obtained by the frequency sweeping method is slightly higher than that obtained by the hammering method.

**Keywords:** modal analysis; stator; motor shell; frequency response function; low-order modal

## 1. Introduction

Motors are used in a variety of dynamic applications. Their vibration and noise have always been research focus [1,2]. The acquisition of the natural frequency and vibration modal of the motor stator is very important for the analysis of motor vibration [3]. The low-order radial modal of the motor stator is the focus of motor vibration and noise. On one hand, the shapes of the first 5th order of the radial electromagnetic force wave are similar to that of the motor stator [4,5]. Thus, there is a resonance of the motor stator as the frequency and modal of a certain order of radial electromagnetic force wave are close to that of the stator [6,7]. For both frequency avoidance and modal avoidance, the frequency and modal of radial electromagnetic force wave should be considered from the design of the motor to avoid the modal order of the motor stator. On the other hand, the deformation of the stator core is inversely proportional to the fourth power of electromagnetic force wave order [8]. A higher order of the electromagnetic force wave leads to a smaller variation of the stator core.

The motor stator system is a complex structure. It is difficult to accurately obtain the material characteristics and parameters of the core laminations and windings, which make it difficult to obtain accurate simulation calculation results [1–6]. The experimental modal analysis can not only evaluate the dynamic characteristics, but also helps to identify the

design defects and verify the product quality [9]. Thus, the experimental modal analysis has always been an important research area of motor vibration and noise.

In recent years, the modal analysis of the motor stator has been widely studied. Neves et al. obtained the modal parameters of the mechanical structure of a squirrel-cage induction motor by the modal test [10,11]. The force acting on the stator was evaluated by the local force density method, and the main vibration spectrum components were identified. Cai and Pillay measured the second-order modal of the switched reluctance motor with acceleration sensor [12]. The results showed that the influence of the length of stator lamination on the plane modal frequency can be ignored. Li et al. obtained the modal shape and natural frequency of the induction motor by experimental methods [13]. They studied the influence of installation angle and rotor on modal shape and natural frequency. The results indicated that the number of low-frequency modals and the frequency of the motor increased with the mounting foot and rotor. Xie et al. conducted a radial second-order modal test on a 1.1 KW asynchronous motor [14]. They also studied the influence of winding, rotor, end cover and casing on the natural frequency of stator. The test prototype included the stator without winding, the stator with winding, the stator without end winding, and the whole stator. The natural frequency of the stator with winding was slightly higher than that of the stator without winding. The second-order natural frequency of the stator was obviously reduced after the winding was dipped in paint, while the end winding had little effect on the natural frequency of the stator. Liu carried out a modal test on a motor stator [15]. The stator system consisted of a stator core, a frame, and a winding. The modal test of the motor system was carried out by the hammering method, and the 1-4 order modal natural frequencies of the stator system were obtained, which were close to the simulation results. Yu et al. presents a modal analysis of stator in large capacity permanent magnet motor [16]. The results of modal analysis indicated that the laminations and windings have a great effect on the natural frequency and vibration shape of the motor stator. Liu et al. presented a modal analysis on a novel modal-independent ultrasonic motor [17]. The modal test showed that the disparity between the modal frequencies of the stators is 0.8%. By arranging the location of rotor and stators, the stators were excited the third longitudinal vibration mode. Lei and Qian conducted a modal analysis on a high-thrust linear ultrasonic motor [18]. The results showed that the inherent frequency of the stator increased with the order and corresponded with the vibration mode. Yan et al. studied the stator modal shapes and natural frequencies of a common Y series induction motor [19]. They provided a method for the motor design by predicting the natural frequency and optimizing the motor parameters to avoid resonance. Qiu presented a modal analysis of the motor base and compared the natural frequencies in different boundary constraints [20]. The motor base was optimized by contrasting different effects on the natural frequencies under different thicknesses of the ribbed plates.

The hammering method was widely used to test the low-order radial modal of the motor stator in the above literature. The test method is relatively simple, and there is no in-depth study on the low-order radial modal test method of the motor stator. The hammering method has limited excitation energy and low signal-to-noise ratio, and the data quality is not too high. It is only suitable for linear structures. Compared with the hammering method, the excitation energy of the frequency sweeping method is larger, the distribution is more uniform, and the data quality is higher. It is suitable for nonlinear structures. In order to deeply test the low-order radial modal of the motor stator, this study applies both the hammering method and the frequency sweeping method to test and analyze the modal of a driving motor. The linear structure of the motor stator is detected, and different modal test methods of the motor stator are compared. The modal frequency, modal shape, and modal damping ratio of the motor stator are obtained, which lays the foundation for an accurate calculation of electromagnetic vibration response.

The structural frame of the stator in the powertrain is a cylindrical shell. Thus, its vibration characteristics cannot completely represent the vibration characteristics of the stator in working conditions. In the previous research on motor vibration and noise

reduction, more efforts were made to improve the motor structure to keep its natural frequency away from the excitation frequency of electromagnetic force, thereby avoiding the occurrence of resonance. However, the stator and stator shell may be developed by different manufacturers. Therefore, it is necessary to discuss the vibration and noise levels of stators with and without shell. Another work of this paper is to compare the natural frequency and vibration modal of the stator with and without shell, and to study the effect of the stator shell on the vibration characteristics of the stator system. This provides a reference for optimizing the vibration characteristics of the stator shell.

## 2. Test Objects and Tools

### 2.1. Motor Stator

The test object is a driving motor stator of the plug-in hybrid electric vehicle, as shown in Figure 1. The windings and stator weigh 30.83 kg and 13.68 kg. The inner and outer diameters of the stator are 230 mm and 290 mm, and the axial length is 77 mm.

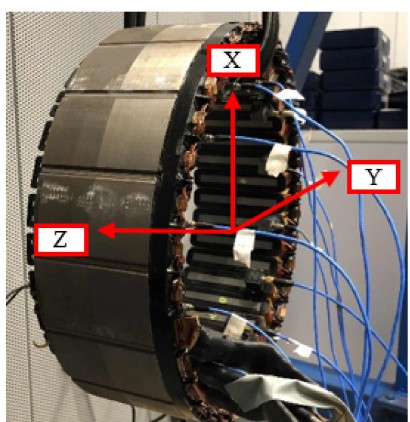

**Figure 1.** Motor stator and its reference coordinates.

### 2.2. Modal Test System

The modal test system is mainly composed of three parts: the excitation source, the response measurement system, and the data analysis system.

Either the force hammer or the vibration exciter is used as the excitation source. The sensitivity of the hammer is 10 mV/lb. The frequency response range is 0 to 8000 Hz. As shown in Figure 2, the electric power modal exciter produced by Tira Co., Ltd. from Schalkau city, Germany is used. The main technical parameters include: a frequency range of 2 to 7000 Hz, main resonance frequency > 5700 Hz, sinusoidal peak thrust 200 N, maximum sine peak velocity of 1.5 m/s, maximum acceleration of 100 g, suspension stiffness of 5 N/mm, maximum payload of 3 kg, total mass of 36 kg, and an armature diameter of 60 mm. The alternating current is connected into the moving coil. The moving coil is driven by the periodic electromagnetic excitation force, which drives reciprocating movements of the pin. The expected vibration can be obtained by contacting the pin with the excited part.

The data acquisition terminal is used to collect the test data, which completes the signal modulation, the A/D (analog/digital) conversion and the digital signal processing. DC (direct current), AC (alternating current), and ICP (integrated circuits piezoelectric) coupling modes are supported. The coupling frequency range of AC and ICP is 0.5 Hz to 23 kHz. Each signal input channel has the function of analog amplification and filtering modulation. The signal input interface is BNC (Bayonet Neill–Concelman) interface. Each channel has an independent A/D converter, which realizes synchronous sampling of each channel. The data acquisition terminal has the function of overload inspection of each channel, which can display and record the overload of each channel. Data transmission adopts universal interfaces such as a 1000 Mb network card and USB (universal serial

bus). The data transmission speed from the data acquisition terminal to host meets the requirements of real-time test transmission. In addition, the data acquisition terminal also has the advantages of anti-electric shock, anti-electromagnetic interference and low noise cooling. The normal temperature acceleration sensors are used in the test. Its range is $\pm$ 500 g. The sensitivity is 10 mV/g. The frequency response is 1 to 5000 Hz. The system accessories include cables and connectors.

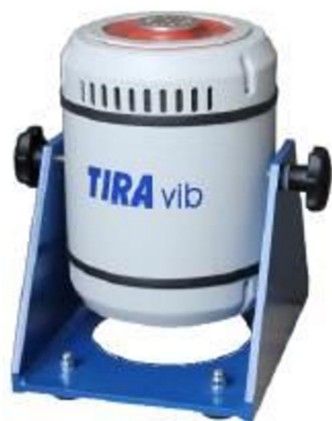

**Figure 2.** Modal exciter of German TIRA.

The Impact Testing software is used as the data acquisition and analysis processing tools for the hammering method; the MIMO (Multiple Input Multiple Output) Sweep & Stepped Sine Testing software is used for the frequency sweeping method.

## 3. Test 1: Effect of Stator Shell on the Stator Modal

Most of previous tests focused on the stator. The modal test of a stator with shell can better reflect its vibration characteristics under working conditions. The well-founded design of the shell is beneficial to the optimization of motor NVH (Noise, Vibration, and Harshness). Therefore, one work of this paper is the modal analysis on the stator with or without shell.

### 3.1. Sensor Layout and Parameter Setting

The Cartesian coordinate system is used to establish the geometric model in this test. It is specified that + X axis represents the stator center to its surface, and + Z axis denotes the axial direction. The + Y axis is determined by the right-hand rule, as shown in Figure 1.

The motor is divided into three sections along the axial direction of the stator. Each section is evenly distributed with 12 measuring points on the inside and outside of the stator along the circumferential direction. The double-root modal may exist due to the asymmetrical stator structure of the motor. In order to extract the multiple root modals, several excitation points are used in the experiment. Each section is set with one excitation point. The electromagnetic vibration and noise is excited due to the radial electromagnetic force wave acting on the stator of the motor. In order to consider the radial modal where the axial order is zero, the excitation direction is fixed to the − X direction. To simulate the free-free boundary condition, the motor stator is suspended on the hanger with a soft elastic rope and kept horizontal. Each test was conducted five times. The test is carried out by moving sensors. Each test is arranged with 12 acceleration sensors. In order to increase the data reliability, it is necessary to check the data quality after each test. If the data quality is good, the sensors are moved to another section. If the data quality is not high, the test should be repeated until the satisfactory data is obtained.

In order to better excite the low frequency vibration signal of the stator, it is necessary to change the hammer head or increase the counterweight. A force hammer with a rubber head without a counterweight was used. The bandwidth, the spectral line, the sampling

time, and the resolution was set to be 3200 Hz, 1024, 0.32 s, and 3.125 Hz, respectively. The acceleration response signal was fully attenuated; thus, no window function is added.

### 3.2. Results and Analysis

The pre-tests such as the signal leakage check test, the frequency response function check test, and the coherence function check test are carried out before the hammering test. Figure 3 shows the coherence function curves of the stator at a certain excitation response point. It can be seen that the values of most coherence functions are higher than 0.95. It shows that the quality of the frequency response function is high, which reflects that the response of the motor stator is completely caused by the excitation source. Compared with the motor stator with shell, the coherence function of the stator without shell is higher, which indicates that the stator without shell is easier to excite its modals.

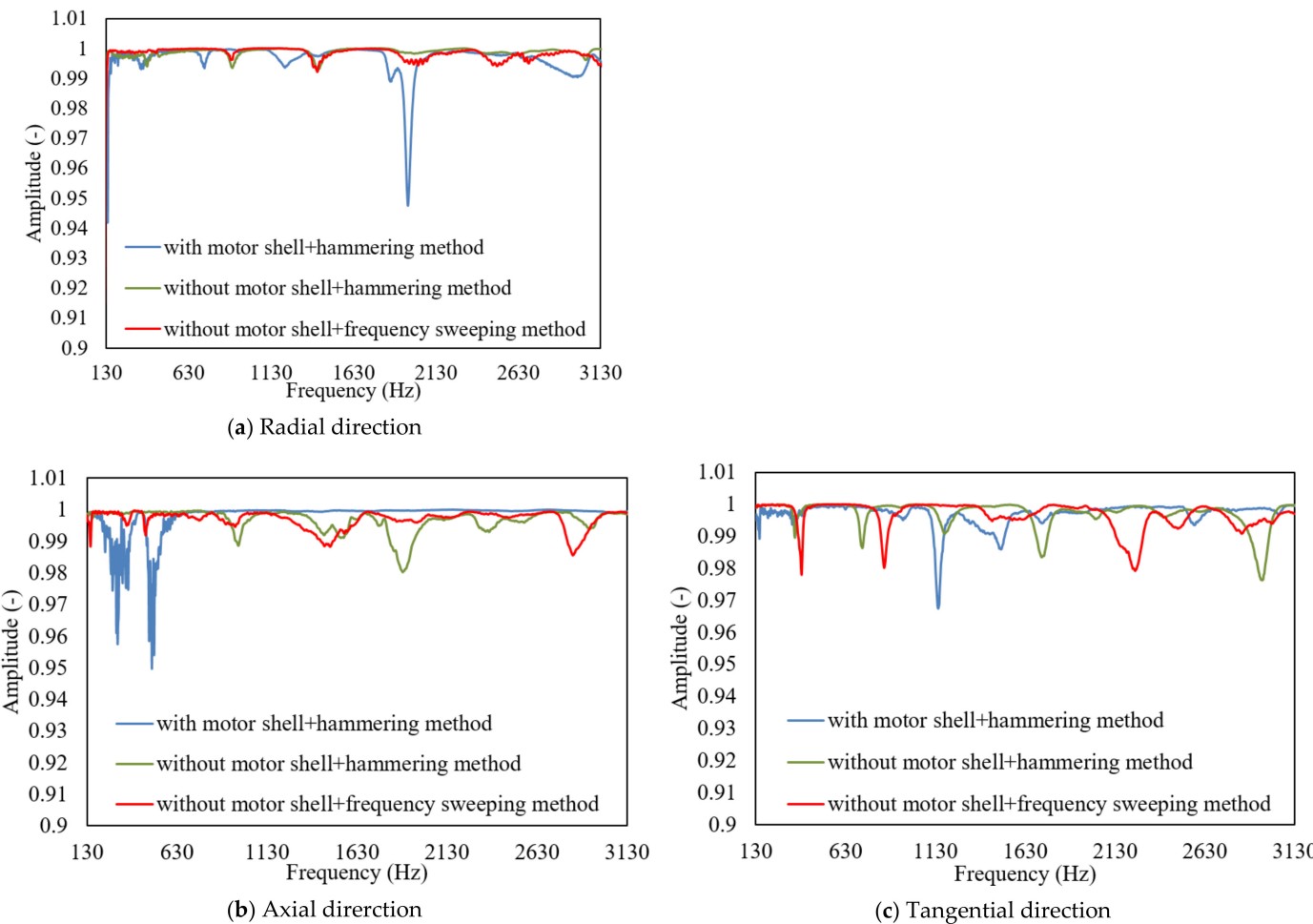

**Figure 3.** Coherence function curves at a certain stimulus response point.

Figure 4 further shows the acceleration frequency response curves in the motor stator modal test. It can be seen that the peak of the frequency response curve of the motor without shell is significant, and its value is obviously higher than that of the motor with shell.

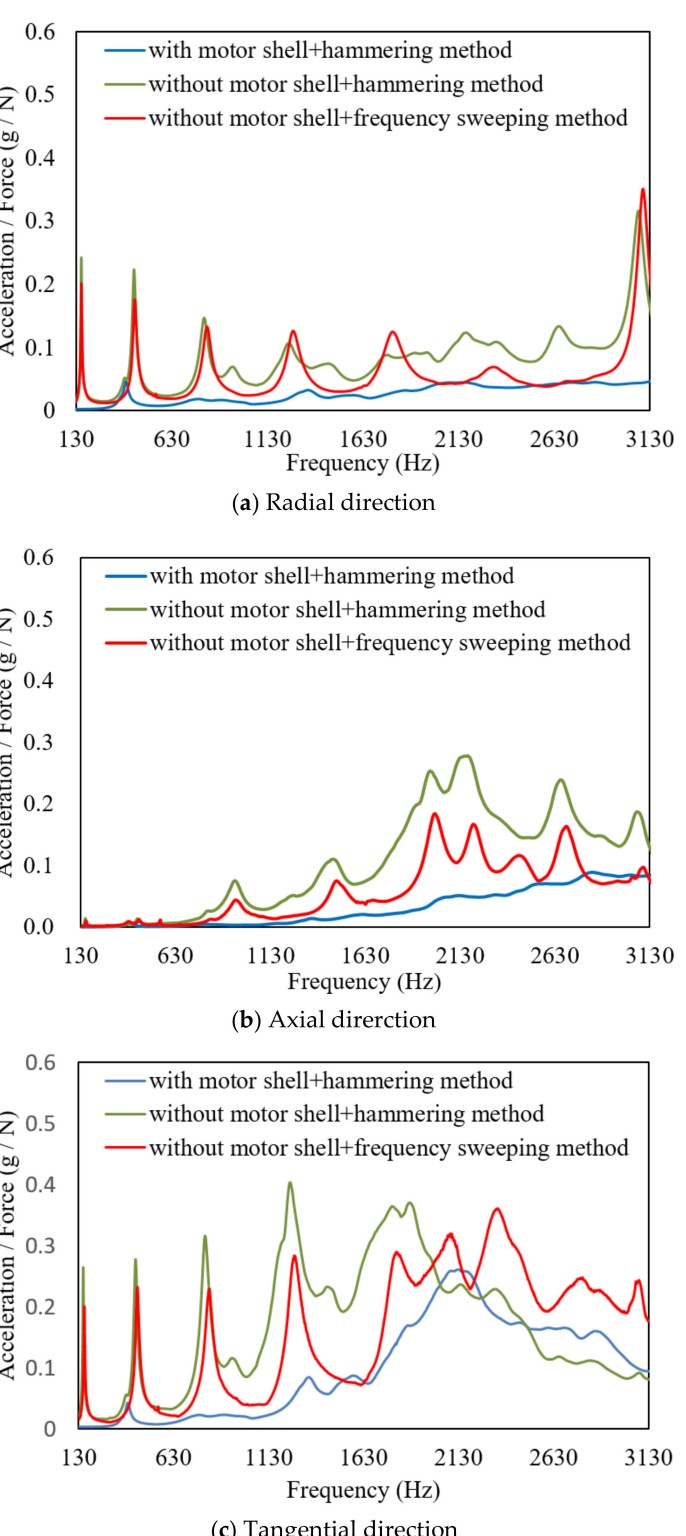

**Figure 4.** Acceleration frequency-response curves of motor stators.

The MIMO modal analysis algorithm is used to analyze the test data. Table 1 shows the damping ratio of modal frequencies and responses sorted by the spatial order. $m$ in $(m, n)$-order modal represents the number of axial nodes of the motor stator, and $n$ is half of the number of radial nodes of the motor stator. $n$ is used to describe the spatial order of the modals. The table shows that the excited modal frequencies of motor stator with shell are obviously higher than those without shell. Optimizing the structure of the stator

shell, choosing steel instead of aluminum alloy as the shell material, and thickening the reinforcement ribs of the stator shell can improve the NVH performance of the stator shell. Stiffness is the main factor that affects the stator's NVH performance. Greater stiffness leads to less vibration and noise. Although the electromagnetic force does not directly act on the stator shell, the shell stiffness also affects the stator's NVH performance. For the motor stator with shell, the modals of (1, 1), (1, 2), and (1, 3)-orders are not excited. The frequency of each order modal is only affected by the weight and stiffness distribution of the structure. The stator weight of the motor with shell is larger, and the stiffness distribution is also different from the motor stator without shell. It is worth noting that the modal frequencies of each order are not sorted from small to large values. The (0, 0)-order is the zero-order modal (also called the breathing modal). Its modal is enlarged or reduced along the radial direction. $n = 1$ to 5 corresponds to the first to fifth modal, respectively. Their modals are shown in Figure 5.

**Table 1.** Modal frequency and corresponding damping ratio for each order.

| Order | Hammering Method (with Shell) | | Hammering Method (without Shell) | | Frequency Sweeping Method (without Shell) | |
|---|---|---|---|---|---|---|
| | Frequency (Hz) | Damping Ratio | Frequency (Hz) | Damping Ratio | Frequency (Hz) | Damping Ratio |
| (0, 0) | 3285 | 2.16% | 3073 | 1.04% | 3087 | 0.81% |
| (0, 1) | 4707 | 2.13% | 2176 | 1.38% | 2189 | 2.14% |
| (1, 1) | - | - | 2665 | 2.13% | 2681 | 1.41% |
| (0, 2) | 388 | 2.27% | 156 | 1.25% | 159 | 1.58% |
| (1, 2) | - | - | 370 | 2.68% | 381 | 2.21% |
| (0, 3) | 790 | 4.52% | 432 | 1.50% | 438 | 2.05% |
| (1, 3) | - | - | 938 | 2.76% | 957 | 3.21% |
| (0, 4) | 1311 | 4.99% | 798 | 1.93% | 815 | 2.40% |
| (0, 5) | 1972 | 2.99% | 1244 | 0.88% | 1263 | 2.68% |

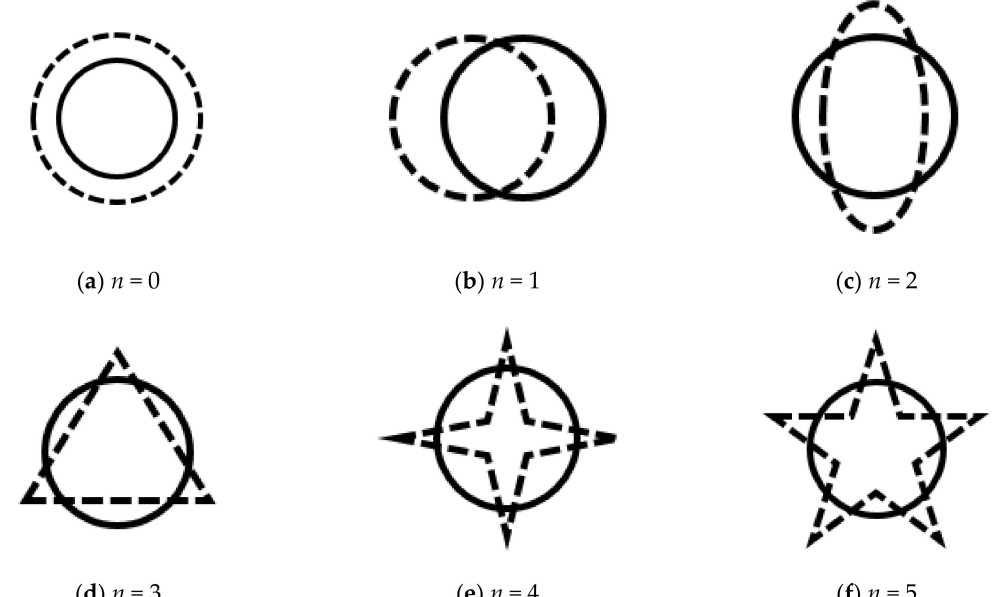

(**a**) $n = 0$        (**b**) $n = 1$        (**c**) $n = 2$

(**d**) $n = 3$        (**e**) $n = 4$        (**f**) $n = 5$

**Figure 5.** Schematic diagrams of modal shapes of motor stator ($m = 0$).

## 4. Test 2: Stator Modal Analysis under Different Excitation Sources

Although the hammering method has many disadvantages, it is more common in practical applications than the frequency sweeping method. Only the frequency sweeping method can be used for excitation as the structure is nonlinear. In addition, the nonlinearity

can be averaged out by selecting the appropriate exciter signal. Another work of this paper is to compare the test results of the modal test on the stator by the hammering method and the frequency sweeping method.

### 4.1. Sensor Layout and Parameter Setting

The modal exciter is placed on the horizontal ground. The bar is vertically fixed on the base, and the force sensor is installed on the outside of the motor stator, as shown in Figure 6. Selecting the excitation point, the node of each order's modal is avoided, and the influence of the excitation point on the exciter and ejector is considered. The MIMO sine modal is set as 'swept'. The control strategy is set to be 'no control'. The frequency sweeping range, the frequency resolution, the sweeping rate, and the frequency sweeping number are set to be 100 to 4000 Hz, 1 Hz, 5 Hz/s, and two times, respectively.

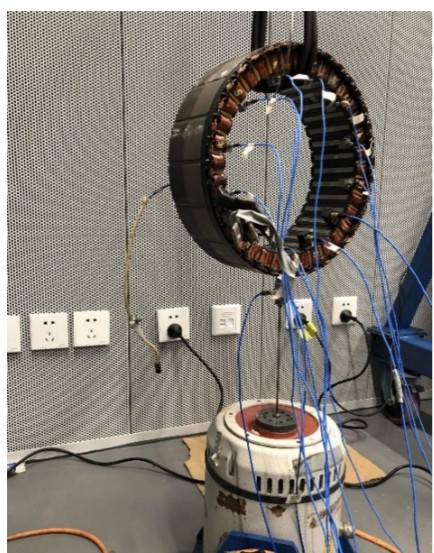

**Figure 6.** Installation of the modal exciter.

### 4.2. Results and Analysis

Figure 3 also presents the coherence function curves of the motor stator under different excitation sources. It can be seen that their values are relatively close. The force hammer with a rubber head and without a counterweight was selected to meet the test requirements, which can stimulate the modals in the frequency range concerned.

Figure 4 also shows the acceleration frequency response curves under different excitation sources. For the motor stator without shell, the consistency of the frequency response function by the hammering method and the frequency sweeping method is better at frequencies below 1000 Hz. The consistency of radial and axial frequency response functions of the two methods is better than that of the tangential frequency response function. Both the exciter and the response sensor have a certain influence on the system structure. The bending stiffness caused by the top rod of the exciter may also lead to errors; thus, the frequency response function obtained by different excitation methods is different.

The linear structure of the motor stator is also proved by the frequency sweeping method. Figure 7 shows the three-dimensional acceleration frequency response curves of the motor without shell obtained by the frequency sweeping method. The exciting force is changed by adjusting the output current and voltage of the exciter. Under different output currents and voltage, the frequency response function of the same excitation response point in all directions is consistent, which indicates that the driving motor stator without shell is linear in the frequency domain concerned. Both the frequency sweeping method and the hammering method are suitable for modal testing of the motor stator without shell.

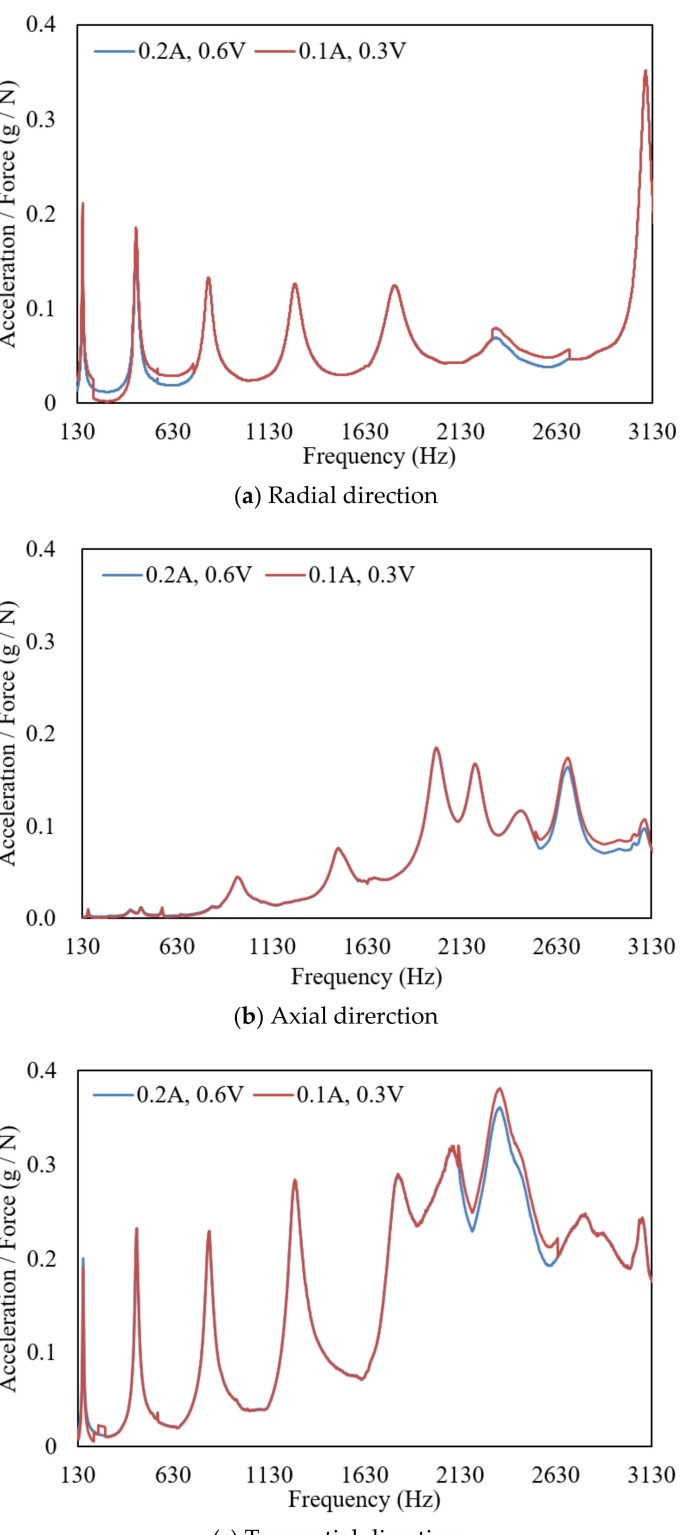

**Figure 7.** FRF curves obtained by the frequency sweeping method.

From Table 1 we can see that, for the motor stator without shell, the frequency of each modal obtained by the frequency sweeping method is slightly higher than that obtained by the hammering method. Their damping ratios are close. The relative error of modal frequency of different excitation sources is 0.45 to 2.88%. There is little difference between the modal frequencies of the motor stator without shell obtained by the hammering method and the frequency sweeping method, respectively.

Due to the axial symmetry of the stator without shell, there exists a multiple root. In addition to the axial nodeless modal ($m = 0$), there are also modals with one node in the axial direction, such as modals (1, 1), (1, 2) and (1, 3). As the number of radial nodes is the same, the larger number of axial nodes lead to a greater modal frequency. Since some modals need to be displayed dynamically, Figure 8 only shows some modals of stator without shell. It can be seen from the figure that the modals of (0, 2) and (1, 2)-orders, (0, 3) and (1, 3)-orders are multiple root modals. In addition, the (0, 4) and (0, 5)-order's modals are also given. The frequencies corresponding to the theoretically multiple root modals are identical. The compactness of the stator is uneven. The three-phase winding busbar of the stator is led out from one side, and the sensors with a certain weight are arranged. To a certain extent, the axial symmetry of the stator is destroyed, and the frequency difference of the multiple root modal frequency is as high as several hundred Hertz. Although their modal frequencies are different, their modal shapes are the same. The modal shapes rotate around the axial coordinates by a certain angle.

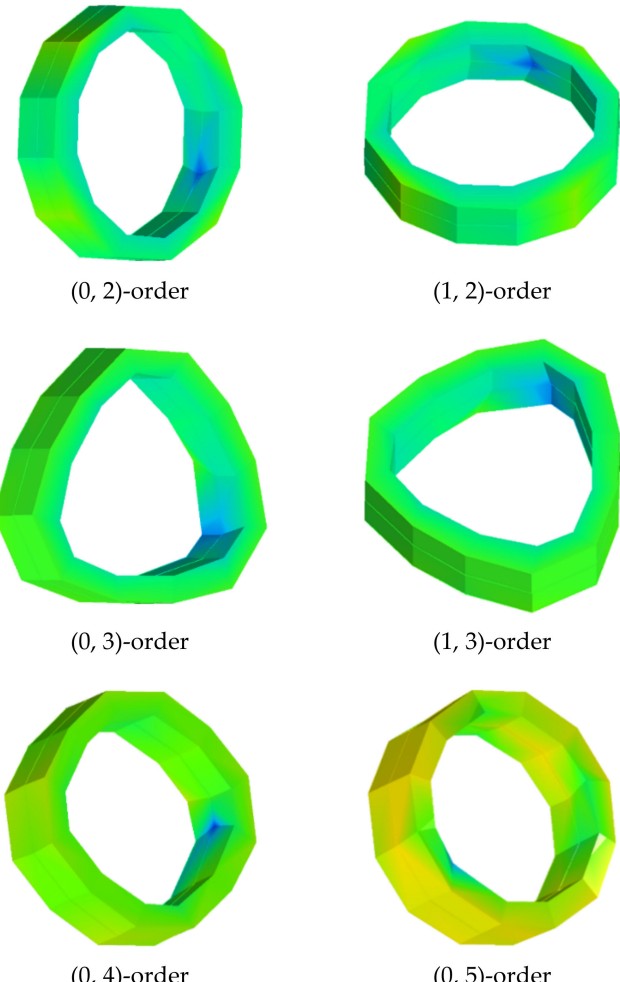

(0, 2)-order           (1, 2)-order

(0, 3)-order           (1, 3)-order

(0, 4)-order           (0, 5)-order

**Figure 8.** Modal shapes of motor stator without shell.

## 5. Conclusions

This study takes a motor stator of a hybrid electric vehicle as the object. The test methods of low-order radial modals of the motor stator with and without shell are studied by using the hammering method and the frequency sweeping method. The results are obtained as follows:

(1) The coherence function of motor stator without shell is higher than that with shell, which indicates that the experimental modal of motor stator without shell is easier to

be excited. The frequency response curve of motor stator with shell is lower than that without shell. The modal frequencies excited with shell are significantly higher than those without shell.

(2)  Under different output currents and voltages, the same excitation response point obtained by frequency sweeping method has the same frequency response function. The motor stator presents a linear structure in the frequency domain concerned. The frequency response functions of the motor stator obtained by the hammering method and frequency sweeping method have good consistency in the low frequency band. The consistency of the radial and axial frequency response functions is better than that of the tangential frequency response functions.

(3)  The (1, 1), (1, 2) and (1, 3) modals of the motor stator with shell can not be excited by the hammering method. The modal frequencies obtained by the frequency sweeping method are slightly higher than those obtained by the hammering method. The motor stator without shell has multiple root modals. As the number of radial nodes is the same, a larger number of axial nodes results in a higher modal frequency.

(4)  The natural frequency and vibration mode of the motor stator are not determined by the stator alone. The stator shell also plays an important role. The NVH performance of the stator cannot be used for evaluating the NVH performance of the stator system, as the impact of the stator shell on the stator system NVH should also be considered.

**Author Contributions:** Conceptualization, S.Y., J.Y., Q.Z. and F.L.; methodology, J.L. and J.S.; software, C.C.; validation, N.W.; formal analysis, Y.C.; investigation, J.L.; data curation, J.L.; writing—original draft preparation, J.L.; writing—review and editing, K.L.; supervision, S.Y., K.L.; funding acquisition, J.L. All authors have read and agreed to the published version of the manuscript.

**Funding:** This research was funded by the Project funded by China Postdoctoral Science Foundation, grant number 2019M663888XB and Natural Science Foundation of Chongqing, China, grant number cstc2020jcyj-msxmX0121.

**Institutional Review Board Statement:** Not applicable.

**Informed Consent Statement:** Not applicable.

**Conflicts of Interest:** The authors declare no conflict of interest.

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
