# Peer review of "Low-Order Radial Modal Test and Analysis of Drive Motor Stator"

_machines, doi:10.3390/machines9050097_

Round 1

Reviewer 1 Report

The authors use the hammering method and the frequency sweeping method to conduct low-order radial modal test of the drive motor stator.

The results are acceptable, while the paper is lack of promising novelty.

Literature review section should be extended considerablely, to show systematic comprehension of authors to the research background. In which, previous research results and methodologies should be discussed thoroughly.

Research significance should be clarified, from methodology improvement concern or practical application oriented, to avoid over research on trial part.

Technology innovations or new understanding of constraints on the analysis of stator structure should be emphasized in the paper. 

Reviewer 2 Report

With the shell, the stator parts are constrained by the shell and have been converted from the  free mode to the constrained mode. The presence or absence of a shell has an effect on the weight of the measured object and also on the frequency response. The frequency and mode shapes of the stator naturally change a lot, which should be considered in the design process of the shell. Therefore, the research content and methods of this paper need to be modified, and it is  suggested to reorganize the thoughts and write it again.

Reviewer 3 Report

  • please explain all abbreviations
  • I recommend using shorter sentences. For example, the first sentence in the abstract is difficult to follow and could have been divided into two different sentences.
  • pay attention to the English language -  for example  the following sentence seems translated ad litteram "The force hammer and vibration exciter are used as the excitation source, respectively."
  • you say that "The cylindrical coordinate system is more suitable to establish the geometric modall (without the last l)  as the stator is axisymmetric" and you use X,Y,Z instead of (ρ, φ, z). Also maybe you can improve figure 1 
  • the figure and its caption must be on the same page
  • I believe that is important to improve the charts quality in some of the figures (they are unclear, there are gaps between the values, portions with red dot lines and portions where the red line is continuous etc. - for example see fig 4 and 5)
  • row 199 and 200 contains a double definition of n
  • I believe that the references can be improved and newer references can be added

Round 2

Reviewer 2 Report

This paper has a clear idea after modification, and it is recommended to accept and publish it.